# Roles of Phosphate Solubilizing Microorganisms from Managing Soil Phosphorus Deficiency to Mediating Biogeochemical P Cycle

**DOI:** 10.3390/biology10020158

**Published:** 2021-02-17

**Authors:** Jiang Tian, Fei Ge, Dayi Zhang, Songqiang Deng, Xingwang Liu

**Affiliations:** 1Department of Chemical Engineering, Xiangtan University, Xiangtan 411105, China; tianjiangjames23@xtu.edu.cn; 2Department of Environmental Science and Engineering, College of Environment and Resources, Xiangtan University, Xiangtan 411105, China; gefei@xtu.edu.cn; 3School of Environment, Tsinghua University, Beijing 100084, China; zhangdayi@tsinghua.edu.cn; 4Research Institute for Environmental Innovation (Tsinghua–Suzhou), Suzhou 215163, China; izzydeng1987@163.com

**Keywords:** phosphate solubilizing microorganisms, soil P, P forms, P biogeochemical cycle

## Abstract

**Simple Summary:**

Phosphate solubilizing microorganisms (PSMs), a large microflora that mediate bioavailable soil P, play a critical role in soil by mineralizing organic P, solubilizing inorganic P minerals, and storing large amounts of P in biomass. Given that the basic soil P forms and orthophosphate levels can be mediated by PSMs, we conclude that PSMs also play a critical role in the soil P cycle. The present review summarizes the comprehensive and recent understanding about the roles of PSMs in P geochemical processes.

**Abstract:**

Phosphorus (P) is a vital element in biological molecules, and one of the main limiting elements for biomass production as plant-available P represents only a small fraction of total soil P. Increasing global food demand and modern agricultural consumption of P fertilizers could lead to excessive inputs of inorganic P in intensively managed croplands, consequently rising P losses and ongoing eutrophication of surface waters. Despite phosphate solubilizing microorganisms (PSMs) are widely accepted as eco-friendly P fertilizers for increasing agricultural productivity, a comprehensive and deeper understanding of the role of PSMs in P geochemical processes for managing P deficiency has received inadequate attention. In this review, we summarize the basic P forms and their geochemical and biological cycles in soil systems, how PSMs mediate soil P biogeochemical cycles, and the metabolic and enzymatic mechanisms behind these processes. We also highlight the important roles of PSMs in the biogeochemical P cycle and provide perspectives on several environmental issues to prioritize in future PSM applications.

## 1. Introduction

Phosphorus (P) is a macronutrient that plays essential roles in plant growth and participates in many metabolic reactions [1]. It is a vital element for life as it is present in biological molecules, including nucleic acids, co–enzymes, phosphoproteins, and phospholipids [2,3,4]. Additionally, P is one of the main limiting elements for biomass production in terrestrial ecosystems and the reason for the ongoing eutrophication of continental and coastal waters because of extensive utilization of P–fertilizers [5,6,7]. The P cycle exists within individual ecosystems including soil, stream, forest, and marine, which are closely related to key security issues of surrounding environment and human society [6,8,9,10]. 

P cycle differs from the N and C biogeochemical cycles since it does not form any stable gaseous species at Earth temperatures and atmospheric pressures [11,12]. Only small amounts of phosphoric acid (H_3_PO_4_) may enter the atmosphere and contribute to acid rain in some cases [13]. P emitted by combustion of fossil fuels and biofuels into the atmosphere, which has been listed as one of the ten critical ‘planetary boundaries’ of the Earth system [14], will be subsequently transported to aerosol-borne P and rapidly deposited in terrestrial and aquatic ecosystems. Thus, the largest reservoirs of P in soil and marine environment are phosphate rock (PR) and sedimentary rock, respectively. The natural P cycle is a simple one-way flow from terrestrial to aquatic ecosystems through inorganic PR denudation and sediment immobilization on medium-term timescales (10^3^ years) [8,15]. 

Previous studies have considered biological activity, human perturbation, and climate change as important factors impacting global P cycling by increasing P concentrations from both external and internal sources, with consequences for terrestrial and aquatic ecosystems [6,8,16,17,18]. Human activities, including the development and utilization of organophosphorus chemicals, extraction of geological P reserves to produce P–fertilizers, and the disposal of animal excreta into the environment, have dramatically impaired soil P geochemical balances and ecosystem functions [19,20,21,22]. Both precipitation and soil temperature have contrasting effects on P availability and control the soil P cycle through interactions with soil particles [23]. Hence, environmental factors involved in soil P geochemical reactions or affecting P-containing compounds can definitely influence the soil P cycle. Although the soil carbon (C) and nitrogen (N) cycles have received considerable attention, much less is known about the change in the P cycle and P availability in response of biological activities and climate change [23].

Phosphate solubilizing microorganisms (PSMs), a large microflora that mediate bioavailable soil P, play a critical role in the soil P cycle by mineralizing organic P, solubilizing inorganic P minerals, and storing large amounts of P in biomass [24,25]. By releasing phosphatase enzymes and organic acids, reducing soil pH, and increasing chelation activities with additional P adsorption sites, PSMs can dissolve soil P into soluble and plant available orthophosphate forms (mostly PO_4_^3–^, HPO_4_^2–^, and H_2_PO_4_^–^) [1]. Therefore, distinguishing the contributions of PSMs to plant nutrition acquisition, understanding the opportunities for manipulating PSMs to enhance soil P availability with the aim of restoring other soil elements, and improving soil health have received considerable interest [1,26]. Although inoculation with PSMs is a widely accepted environmentally-friendly approach for increasing soil soluble P concentrations and agricultural productivity, a comprehensive and complete understanding about the roles of PSM in P geochemical processes (i.e., dissolution–precipitation, sorption–desorption, and weathering) has not received the merited attentions [27]. In addition, the efficient utilization of PSMs in situ remains in its infancy as the wide-ranging applicability and potential eco-toxicity has not yet been demonstrated. In this review, we discuss the basic P forms and P cycling in the soil, how PSMs mediate soil orthophosphate levels, and the biogenic mechanisms behind these activities. Finally, we highlight the roles of PSMs in each P biogeochemical process, and propose environmental issues to prioritize in future PSM applications.

## 2. Basic P Forms and P Cycling in the Soil

Most soils contain higher levels of total P (P_t_) than other major nutrients, such as nitrogen (N) and potassium (K). However, more than 80% of P is immobile and not readily accessible for plant uptake [28]. P exists in different forms in soil, mainly inorganic P (P_i_) and organic P (P_o_) (Figure 1), and the proportions of P_i_ and P_o_ change as soils develop [29].

Soil P_o_, which contributes a large proportion of total soil P, originates mainly from biological tissues where P is present as an integral part of organic compounds, such as nucleotides, phospholipids, phosphoproteins, and co-enzymes [1,30]. Soil nutrient cycling processes (e.g., nitrogen cycling) are also responsible for the re-distribution of primary P_i_ into P_o_ forms over timescales of 10^4^–10^6^ years [30]. The widespread application of P_o_-containing products, such as plasticizers, fire retardants, antifoam agents, and pesticides, has resulted in their frequent occurrence in the environment as new P_o_ sources, consequently increasing the quantities and varieties of P_o_ forms in soils [31,32,33]. 

Unlike P_o_, which is more easily leached because of weak interactions with the soil particles [34], soil P_i_ is usually present as relatively insoluble and stable forms of primary (including apatite, strengite, and variscite) and secondary (including calcium, iron, and aluminum phosphates) P minerals [35,36,37,38]. Early studies developed chemical fractionation schemes to determine the fractions of P_i_ as (i) acid-extractable calcium-bonded P (including apatite and lattice-P), (ii) non–occluded P (including NH_4_F and NH_4_Cl extractable P, and 1st NaOH extractable P), and (iii) occluded P (including 2nd NaOH extractable P and residual P_i_) [39,40]. The P ions in non-occluded P are adsorbed at the surfaces of Fe and Al oxides and are more easily extracted by NaOH than occluded P, in which P is incorporated within developing Fe and Al oxide coatings and concretions during diffusive penetration and soil development [39]. P_i_ concentrations also decrease as a function of soil development, ranging from an average of 684 µg /g soil in Entisols, to between 200 and 430 µg/g soil in Ultisols and Oxisols [29].

P_i_ exists in different forms and proportions in soil, which may leach into streams to deposit P in ocean sediments, or be taken up by plants or soil microorganisms in the secondary P_o_ cycle [41]. After mineralization by plant structures (e.g., roots, stems, leaves), dead soil animals, and microorganisms, a large share of the assimilated P_o_ returns to soils [42]. Even though P has a number of indispensable biochemical roles in the soil, it does not have rapid cycles compared with those of C, N, and S, which are transported not only in soil and water but also in the atmosphere [42,43]. In most natural ecosystems, geochemical processes, including weathering, adsorption/desorption, precipitation/dissolution, and solid–phase transformations (Figure 1), determine the forms (available or unavailable to plants) and distribution of P in soils over long-term timescales (> 10^3^ years) [23]. However, in the short-term (10^–2^ to 10^0^ years), biological processes influence P distribution because most of the plant-available P derived from soil organic matter is immobilized and mineralized by soil microbes [2,29]. While the effects of geochemical processes in controlling soil P availability are rather well understood, much about the importance of biological processes remains unknown [2]. 

## 3. PSM Enhance Soil P Cycle through Organic P Mineralization

Soil P_o_, which derives mainly from biomolecules, including nucleotides, phosphides, co-enzymes, phosphoproteins, sugar phosphates, and phosphonates, plays an essential role in soil P cycling [2,3,4]. P_o_ substances (e.g., orthophosphate esters, phosphonates, and polyphosphates) are mostly short-lived compounds, and may comprise as much as 65% of P_t_ in most soils [31,44]. Based on its sources, soil P_o_ can be considered to exist in a rapidly cycling pool (fast P_o_) and a slowly cycling pool (slow P_o_) [44]. The fast pool consists of the constant P_o_ from soil solution immobilized in microbial biomass and resupplies the slow pool following cell death. Soil soluble orthophosphate ions can be immobilized in microbial cells to improve biomass growth. It is found that most of the P mineralized from organic P by PSM is incorporated into the bacterial cells as cellular P [45]. Concurrently, these soil microbes can rapidly release P_o_ to the slow pool following cell lysis, cell death, and soil fauna predation [44,46]. Plant detritus, dead animals and microbes, and non-living P_o_ fertilizers (e.g., dry straw and animal manure) are the most common slow P_o_ sources that can directly replenish soil orthophosphate contents through geochemical or biological decomposition, beneficial for plant-available P supplies and soil quality improvement (Figure 1) [47,48,49]. Hence, manipulation of the orthophosphate release from soil P_o_ sources is an important soil P cycle, which has the potential to increase the availability of soil P_o_ for plant uptake and reduce reliance on the P fertilizer inputs. Soil microbes, especially PSM, can enhance soil P_o_ cycle through P_o_ mineralization and decomposition. By analyzing soil P pools and oxygen isotope ratios in P (δ^18^O_P_), Bi et al. [50] have uncovered that soil microbes could activate the soil P cycle by promoting extracellular hydrolysis of P_o_ compounds and facilitating the turnover of bioavailable P pools (H_2_O-P_i_, NaHCO_3_-P_i_, and NaOH-P_i_). These biogeochemical processes are mainly moderated by the activities of phosphatase enzymes in PSM and soils [48]. 

PSMs isolated from bulk soils and rhizospheres have been shown to hydrolyze P_o_ by the release of phosphatases [50]. Phosphatases are enzymes responsible for P_o_ decomposition and mineralization by catalyzing the hydrolysis of both esters and anhydrides of phosphoric acid, and they are usually classified as phosphomonoesterases, phosphodiesterases, and enzymes acting on phosphoryl–containing anhydrides or on P–N bonds [51]. These enzymes originate mainly from soil microorganisms and plant cells, and the enzyme activities are always higher in rhizosphere than in bulk soil [47]. The P_o_ hydrolysis activities of extracellular phosphatase enzymes are affected by soil properties, microbial interactions, plant cover, and environmental inhibitors and activators [51]. Some key environmental traits associated with phosphatase enzyme activities can be genetically manipulated by the regulation of P-cycling-related genes in PSMs and other microorganisms under P deficiency conditions [26,52]. Although soil PSM is involved in P_o_ mineralization and P cycle at various scales, the dominant enzymes and the functional genes are always similar. It has been established that the dynamics of microbial genes and expression of phosphatase enzymes are the key factors governing mineralization of P_o_ into bioavailable orthophosphates by PSMs [50]. 

The non-specific acid phosphatases (phosphohydrolases, or NSAPs) released by PSMs perform dephosphorylation of phosphodiester or phosphoanhydride bonds in P_o_ matters, and they play a major role in P_o_ mineralization in most soils [53]. These NSAPs may be either acid or alkaline phosphomonoesterases [54], and several NSAP genes have been isolated and characterized in PSMs [53]. For example, *olp*A gene of *Chryseobacterium meningosepticum* encodes and expresses a broad–spectrum of phosphatases that efficiently hydrolyze monophosphates and sugar phosphates [55]. PhoN and PhoK from indigenous soil PSMs could express periplasmic acid phosphatase and extracellular alkaline phosphatase, in genetically modified *Deinococcus radiodurans* and *Escherichia coli*, respectively, to enhance the biomineralization of toxic ions in polluted soil [56]. Cyanobacterial *Microcystis aeruginosa* harbors genes encoding extracellular alkaline phosphatase to utilize a variety of inorganic or organic insoluble P [57,58]. Phytase enzyme, encoded by *app*A or *phy*A genes, is another important P_o_ mineralization enzyme responsible for P release from phytate in soil [54]. Previous studies have focused on applications of phytase in the P_o_ mineralization as phytate is the major component of P_o_ forms in soil [53]. Approximately 30–48% of culturable soil microorganisms were reported to utilize phytate by producing phytase enzyme [59]. Yeasts (including *Pichia acacia* and *Candida argentea*) were also demonstrated to produce phytase and utilize phytic acid as their sole P source [60], while a great diversity of phytase exists in the vast majority of unculturable soil microorganisms, which have been rarely studied. Using metagenomics, Farias et al. [61] constructed environmental genomic libraries to determine the complete sequencing of the clone phytase gene from unculturable soil microorganisms in red rice crop residues and castor bean cake. The newly isolated phytase enzyme showed high hydrolase activity at neutral pH under β-propeller structure. Therefore, it is crucial to develop and utilize more advanced approaches to support the roles of PSM–derived enzymes in releasing free orthophosphate from organic P forms in the soil P cycle [25,54,59,62].

## 4. PSM-Mediated Inorganic P Solubilization to Enhance Soil Orthophosphate Contents 

P_i_ is an essential but non-renewable resource for plants on Earth. As plant-available P_i_ comes mainly from the soil environment, increasing global food demand has substantially increased the use of PR for the production of P fertilizer [63]. Previous studies have shown that the peak of global PR mining is estimated to occur in 2030, and the global P_i_ mine will be exhausted in the next 50 to 100 years [7,64]. The types of P_i_ in PR and other primary P_i_ minerals are insoluble and unavailable to most plants. In addition, modern agricultural consumption of PR and P_i_ fertilizers can lead to excessive accumulation of soil P in intensively managed croplands, resulting in P losses and eutrophication of surface waters [65]. Thus, sustainable and judicious PR mining management strategies to reduce water pollution and eutrophication problems have attracted worldwide attention [7]. From the last century to the present, researchers have proved that the application of PSMs is an accepted solution for successful PR mining management and agricultural sustainability [66,67,68,69]. 

In early studies, PSMs are defined as microbes that transform insoluble P_i_ and P_o_ to soluble P forms and regulate biogeochemical P cycling in agroecosystems [70,71]. Among heterogeneously distributed soil microflora, a large number of heterotrophic and autotrophic microorganisms, including phosphate solubilizing bacteria (PSB), phosphate solubilizing fungi (PSF), phosphate solubilizing actinomycetes (PSA), and cyanobacteria are commonly identified as PSM [65,72]. PSB, such as *Bacillus* sp., *Pseudomonas* sp., *Rhizobium* sp., and *Escherichia* sp., form the largest microbial communities with P solubilization abilities in soil. PSF have greater P_i_ solubilizing abilities than PSB by attaching P_i_ minerals to increase the contact area in liquid medium [73]. *Penicillium* sp., *Aspergillus* sp., *Mucor* sp., and *Rhizopus* sp. are the commonly isolated and demonstrated PSF microflora in soil. Some previous studies have found that several arbuscular mycorrhizal fungi (AMF), such as *Rhizophagus irregularis* [74], *Glomus aggregatum*, and *Glomus mosseae* [75], can also solubilize P_i_ either directly through exudation, or indirectly through modification of soil PSM communities [74]. Owing to their dominance and strong antimicrobial potential, filamentous actinobacteria have been extensively used for colonizing plant tissue and producing antibiotics, anti–fungal, and phytohormone-like products that can be beneficial to plant growth [62,76,77,78]. Moreover, actinobacteria, especially *Streptomyces* and *Micromonospora*, are of increasing interest since these sporulating bacteria are capable to solubilize insoluble P minerals in P_i_ weathering process and the soil P cycle [78,79,80]. In addition to these common heterotrophic PSM, autotrophic microorganisms with P_i_ solubilizing abilities were also reported [65]. Yandigeri [81] evaluated the rock phosphate and tricalcium phosphate solubilizing abilities of two diazotrophic cyanobacteria, *Westiellopsis prolifica*, and *Anabaena variabilis*. Periodic monitoring showed that both cyanobacterial strains could significantly increase P_t_ and available-P contents in medium in P-starved or insoluble-P cultures. 

PSMs, including PSB, PSF, and PSA, can transform insoluble P_i_ to soluble orthophosphate forms (PO_4_^3–^, HPO_4_^2–^, and H_2_PO_4_^–^) by secreting various organic or inorganic acids that release H^+^ and lower the medium pH [27,72] (Figure 2). Moderately labile P_i_ (P_i_ partly from Fe/Al–P and from the surface of sesquioxides) can become accessible to soil organisms through organic acid excretion by PSMs which, in turn, chelate Fe and Al ions so that P is liberated [82]. The carboxyl groups of organic acids can bind P by replacing cations or compete for P adsorption sites, enhancing the soil absorption of PO_4_^3–^ and increasing P_i_ solubilization. Notable levels of organic acid production and P_i_ solubilization performance are achieved by PSM isolates [83]. Although PSB represent the largest PSM population in soil environment, PSF exhibit greater P_i_ solubilization abilities by producing 10 times more organic acids than PSB, declining pH by 1–2 units in both liquid and solid media [73]. Furthermore, different P_i_ minerals show a range of H^+^ production and P_i_ solubility levels, which can be explained based on *K*_sp_ values, acidity coefficients, and chemical equilibria [83]. Under strong acid conditions, PO_3_^4−^ will be dissolved first from P_i_ minerals and protonated to form hydrogen P (HPO_4_^2−^ or H_2_PO_4_^−^). Metal ions (e.g., Ca^2+^, Fe^3+^, or Al^3+^) are likely to subsequently capture the hydrogen P to form metal hydrogen P with generally higher *K*_sp_ values than their equivalent metal P [84]. Hence, P_i_ minerals can almost completely dissolve under strongly acidic conditions. For example, the majority of Ca_3_(PO_4_)_2_ solubilization occurs in the pH range of 2.5 to 4.0, while FePO_4_ solubilization occurrs from 2.0 to 2.5 [73]. Accordingly, this explains the lower P_i_ solubilization efficiencies obtained with monocarboxylic acids (acetic, formic, lactic, and gluconic acids) compared to di- and tri-carboxylic acids (oxalic, malic, and citric acids), which have higher acidity coefficients [73,84].

H^+^ may originate from other biotic phenomena, for example from H_2_S and H_2_CO_3_ respiratory acidification (Figure 2), but these acidification processes cannot be responsible for all P_i_ solubilization. Instead, the total solubilization of apatite and brushite were observed in H^+^ production accompanying NH_4_^+^ assimilation in *Pseudomonas* sp. and *Penicillium* sp. [85]. Exopolysaccharides (EPS), important polymers consisting mainly of carbohydrates, were suggested as an important factor in P_i_ solubilization by PSB [27]. EPS could disturb the homeostasis of organic acids or H^+^ involved in P_i_ solubilization process by holding free P in the medium, consequently resulting in more P release from P_i_ minerals [86]. However, further studies are required to understand the mechanisms of synergistic P_i_ solubilization by organic acids and EPS.

*Actinobacteria* and *Cyanobacteria* have rarely been reported for production and quantification of organic acids in P_i_ solubilization [76,79]. *Streptomyces* sp. were isolated from wheat rhizospheric soil and demonstrated to solubilize Ca_3_(PO_4_)_2_ and PR by malic and gluconic acid secretion in glucose-supplemented medium [76]. PSAs of *Streptomyces* sp. and *Micromonospora* sp. were previously reported to solubilize PR by producing Ca-chelators or siderophores [78]. In addition to organic acid secretion and H^+^ dissolving mechanisms, P may be liberated from P_i_ minerals by the synthesis of chelators (e.g., Ca, Al, and Fe-chelators) in PSB, PSA, and blue green algae (Figure 2) [78,81,87,88]. Since PR and other primary P_i_ minerals are mainly insoluble hydroxyapatite, or Ca, Al, and Fe/phosphate, the siderophores and chelators could form chelates with Ca, Al, and Fe, resulting in the release of P originally bound by these metals [81]. 

## 5. PSM-Derived P Desorption from Clay Minerals 

Soluble P forms are easily fixed and removed from soil solution by adsorption reactions and incorporated into the solid phase. These chemical reactions are particularly strong on the surfaces of amorphous iron (Fe) and aluminum (Al) (hydr)oxides in highly weathered tropical and volcanic ash soils (Figure 1) [89,90]. Fe and Al (hydr)oxides are the most important variable-charge minerals with higher P adsorption capacity and binding energy compared with permanent-charge minerals [91]. After long–term fertilization, most soil P is absorbed to Al (hydr)oxides, whereas, in unfertilized soils, P absorbed on Fe is the dominant P species in clay minerals [92]. The strong correlation between Fe (Al) and orthophosphate produces as Fe–P (Al–P) hydroxides with a high capacity of sorption that results in soil P deposition [93,94]. In addition, soil organic matters can also absorb or immobilize orthophosphate into relatively stable P_o_ compounds and reduce dissolved P concentrations [95]. 

Indigenous soil microorganisms with P solubilization abilities were identified to desorb P from the surfaces of clay minerals and soil organic matter [89]. He and Zhu [91] reported that microbial transformation and desorption of P from the surfaces of variable charge–minerals, including Fe and Al (hydr)oxides, predominates in Chinese red soils. Seventeen to thirty-four percent of clay mineral-absorbed P are transformed and desorbed to water-soluble and plant-available P by microbes. As the capacity of PSM to desorb P depends on the P_i_-adsorption capacity, surface areas of clay minerals, and the saturation of P_i_-absorbing sites, the effectiveness of *Mortiella* sp. to desorb P is ranked as montmorillonite > kaolinite > goethite > allophane [90]. Furthermore, inoculation with *Mortiella* sp. can significantly increase in situ soil P_i_ solution through desorption, and the efficiency is enhanced by increasing the absorbed P_i_ levels in different soils. It can be concluded that the capacity of PSM to augment soil soluble P_i_ is directly related to the quantity of P absorbed by soil and clay minerals, while the magnitude of P_i_ desorption by PSM is inversely correlated with the P_i_-sorption capacities of soil and clay minerals [96]. 

The phenomenon of P desorption by PSM usually occurs along with the drop of pH. It is presumed that P desorption may result from the increased solubility of Fe and Al by the possible complexation with low molecular weight organic acids [91,97]. Plant roots and PSM could release various low molecular weight organic acids, such as citrate, oxalate, and malate, during P_i_ mineral solubilization processes, and these organic acids are widely recognized to enhance P_i_ availability in soils through desorption [44,90,97]. Organic acids and anions can displace P_i_ from absorbing sites through ligand exchange from microbial activities and transitory blockage of P_i_ adsorption sites [98]. Depending on their dissociation properties and carboxylic groups, organic acids can carry varying negative charges to increase the desorption of P_i_ [97]. For example, the effectiveness of organic acids in reducing P_i_ sorption follows the order tricarboxylic > dicarboxylic > monocarboxylic acid, which is explained by the constant of complex formation values [96]. Oburger [99] found that the adsorption and desorption isotherms of organic acids could be described by the Freundlich equation and the dynamic sorption model. This model succeeds in both predicting the solid solution partitioning of citrate in soils and demonstrating the plateau and steady state concentrations of citrate in solution, highlighting the key effects of organic acid dynamics on the P_i_ adsorption-desorption reactions and the functional roles of PSM in soil [98,100].

## 6. PSM-Induced Dissolution in Accelerating Metal Precipitation to form Secondary Pi Minerals

Insoluble P_i_ in soils is always constrained by the presence of Ca, Fe, Al, or heavy metal cations [84]. Generally, in acidic soils, P ions tend to precipitate with Fe and Al cations to form insoluble oxyhydroxides or secondary P_i_ minerals. In alkaline soils, P ions mainly precipitate with Ca to form secondary P_i_ minerals, such as fluroapatite, hydroxyapatite, and chloroapatite [101,102,103]. Thus, the geochemical precipitation of P in soil and wastewater have contrasting effects on physicochemical stabilization of organic P compounds and environmental control of P_i_ levels [44,104,105]. 

The apatite families, including Ca–phosphate apatites (Ca_5_(PO_4_)_3_X, where X = Cl, OH, or F) and strontium–phosphate apatite (Sr_5_(PO_4_)_3_H), are environmentally important secondary minerals [106,107]. PSMs can accelerate apatite formation by the release of P_i_ and the hydrolysis of P_o_ with alkaline phosphataes [108]. An oxygen isotope tracing method reveals that a metastable apatite precursor is initially precipitated and then transforms to hydroxy apatite on the surface of microbial filaments, suggesting that the apatite precipitation process involves extensive biological turnover of P_i_ by microorganisms [109]. Fe–P minerals are effectively precipitated and formed by orthophosphate ions and goethite (α–FeO(OH)), the most common secondary Fe oxyhydroxide in natural environments, through monodentate- or bidentate-complexing process [101]. Accordingly, precipitated Fe–P minerals, including vivianite, strengite, and ferrihydrite, are demonstrated as effective and promising approach to improve P removal and recovery [110]. These precipitated P minerals can release P in soil upon dissolution, the processes of which are enhanced by soil PSMs.

Among Ca–P secondary minerals, hydroxyapatite is a common P source for soil microorganisms and is often utilized to isolate PSMs accompanied by Ca_3_(PO_4_)_2_ and apatite [111,112]. This crystallized Ca–P mineral is much more dissolution–resistant than amorphous Ca_3_(PO_4_)_2_, and hydroxyapatite dissolution is driven by the microbial production of D–gluconic and 2–keto–D–gluconic acids, which were more effective than P_i_ desorption from Fe oxyhydroxide [101]. Fe–P and Al–P minerals also commonly exist as plant P sources in soil. These secondary minerals show relatively higher p*K*_sp_ and lower dissolution rates than Ca–P minerals [85,113], and the dissolution mechanisms are characterized as plant- or PSM-mediated organic acid release (such as citric and piscidic acids that have high acidity coefficients) [114] and iron–binding siderophore production [115]. 

Research has focused increasingly on biomineralization of heavy metals to form stable precipitations or secondary P_i_ minerals in soil [116,117,118,119]. Biomineralization based microbial induced phosphate precipitation (MIPP) is a novel approach for soil heavy metal remediation. By adding exogenous P sources, indigenous PSMs can release various organic acids and phosphatases to increase soluble orthophosphate concentrations, which then mediate metals ions mineralization as P–containing minerals [56]. Many toxic heavy metals including Pb, U, Zn, Cu, and Cd are reportedly immobilized as stable P–containing minerals through MIPP processes [120]. Among them, Pb is the most frequently reported heavy metal that can be precipitated by MIPP (Figure 3). Pb ions can be immobilized by cell surface biosorption, organic acid-mediated precipitation, and P-containing mineralization [117,121]. These P-containing minerals include Pb_3_(PO_4_)_2_, Pb_9_(PO_4_)_6_, lead apatite (Ca_(10–x)_Pb_x_(PO_4_)_6_(OH)_2_), and pyromorphite family (Pb_5_(PO_4_)_3_X, X = Cl^–^, OH^–^, Br^–^, F^–^, *K*_sp_ = 10^–71.6^ ~10^–84.4^), considered as the most stable precipitated Pb minerals in the environment [56,117,122]. 

In the presence of P_i_ minerals (such as apatite and PR), Pb^2+^ can replace Ca^2+^ by ion exchange to form lead apatite (Equation (1)), or be absorbed to form Ca_8_Pb_2_(PO_4_)_6_(OH)_2_ [123]. In the presence of both P_i_ minerals and PSM, hydroxypyromorphite precipitation includes the reaction of Pb^2+^ with apatite through PSM–induced hydroxyapatite dissolution as shown in Equations (2) and (3) [119].
(1)$${\mathrm{Ca}}_{10}{\left({\mathrm{PO}}_{4}\right)}_{6}{\left(\mathrm{OH}\right)}_{2}+{\mathrm{xPb}}^{2+}\overset{\;\;}{\to }{\mathrm{Ca}}_{10-x}{\mathrm{Pb}}_{x}{\left({\mathrm{PO}}_{4}\right)}_{6}{\left(\mathrm{OH}\right)}_{2}$$
(2)$${\mathrm{Ca}}_{10}{\left({\mathrm{PO}}_{4}\right)}_{6}{\left(\mathrm{OH}\right)}_{2}\left(s\right)+{14H}^{+}\left(\mathrm{aq}\right)\overset{\;\mathrm{dissolution}\;}{\to }{10\mathrm{Ca}}^{2+}\left(\mathrm{aq}\right)+{6H}_{2}{\mathrm{PO}}_{4}{}^{-}\left(\mathrm{aq}\right)+{2H}_{2}O$$
(3)$${10\mathrm{Pb}}^{2+}\left(\mathrm{aq}\right)+{6H}_{2}{\mathrm{PO}}_{4}{}^{-}\left(\mathrm{aq}\right)+{2H}_{2}O\overset{\;\mathrm{precipitation}\;}{\to }{2\mathrm{Pb}}_{5}{\left({\mathrm{PO}}_{4}\right)}_{3}\mathrm{OH}\left(s\right)+{14H}^{+}$$

PSBs *Pabtoea ananatis* and *Bacillus thuringiensis* can effectively solubilize PR to release soluble P, which rapidly reacts with Pb^2+^ to form insoluble lead minerals, and reduces the phytoavailability of Pb^2+^ to benefit plant grwoth and net photosynthetic rate [124]. Zhang [118] found that the Pb_3_(PO_4_)_2_ and Pb_5_(PO_4_)_3_OH precipitates produced by PSBs decomposing organophosphorus polymers were more stable than those of urease-producing bacteria that produce PbCO_3_. Furthermore, other toxic metals, such as U, Cu, Zn, Cd, and Ni, have also been reported to be bioimmobilized by MIPP. For example, U could react with PSM–released orthophosphate to precipitate numerous U–P minerals, such as chernikovite (H_2_(UO_2_)_2_(PO_4_)_2_), autunite (Ca(UO_2_)_2_(PO_4_)_2_), uramphite (NH_4_(UO_2_)_2_(PO_4_)_3_H_2_O), and ankoleite (K_2_(UO_2_)_2_(PO_4_)_2_) [56]. PSA *Streptomyces* showed high heavy metal resistance and mineralization properties to form crystallized P–containing switzerite (Mn_3_(PO_4_)_2_•7H_2_O) and hydrated nickel hydrogen phosphate [125]. Therefore, PSM-induced P_i_ mineral dissolution can both supply bioavailable orthophosphate to plants and accelerate metal precipitation to form secondary P-containing minerals in soil. 

## 7. Effect of PSM on Pi Mineral Weathering and the Biogeochemical P Cycle

Chemical or biological weathering of primary P minerals has substantial influence on the global biogeochemical P cycle [36,126,127]. Continental weathering of P minerals provides the ultimate source of bioavailable P to marine systems and supplies almost the entire source of P in most soil profiles [128]. Globally-enhanced continental weathering has delivered vast amounts of P to the oceans, resulting in the increased levels of subsequent eutrophication and marine anoxia [12,36]. The weathering of P minerals depends on numerous environmental factors, including earth history, environmental erosion, atmospheric composition, rock P contents, soil microaggregate fractions, and biological response [15,36,126,129,130,131]. As soil microorganisms with P_i_ mineral weathering abilities, PSM or phosphate-dissolving microorganisms, are environmentally widespread [111,132]. These microorganisms can release orthophosphate from amorphous P_i_ minerals, such as Ca_3_(PO_4_)_2_, FePO_4_, P–bearing mineral powders, and crystalline P_i_ minerals, such as apatite, fluorapatite, and phosphorites, mainly through acidolysis [111,132,133,134]. PSM *Pseudomonas fluorescens* can dissolve fluorapatite as its sole P source to release P, Ca, and F in acidified medium, which is important in the weathering of fluorine-bearing minerals [132]. Several unfamiliar PSMs, along with *Micromonospora* sp. and *Streptomyces* sp., are found to produce siderophores but not organic acids during PR weathering processes [135]. PSMs with thermo-tolerance and drought tolerance abilities are able to release large amounts of other useful minerals, such as K, Mg, Fe, and F, by weathering P minerals, fluorapatite, extrusive igneous rock and limestone [132,136]. Therefore, the net effects of PSMs on P_i_ mineral weathering are similar to those of P_i_ mineral solubilization, resulting in accelerated dissolution of primary P_i_ minerals to release P and laterally transporting P in soil systems. 

## 8. PSM Enhance P Uptake from Soil to Plant in the Rhizosphere Environment

Bioavailable P content in soil is an important factor to enhance plant P uptake and achieve higher crop yields [72,137]. Soil PSM can employ different biogeochemical strategies to make use of the unavailable P forms and in turn help in enhancing P uptake from soil to plants. Hence, most studies have considered PSM as a promising inoculant/biofertilizer for raising the productivity of agronomic crops in agroecological niches [72,137,138]. However, the soil is a more diverse and spatially heterogeneous matrix than a growth medium, which will result in some of the discrepancies between the in vitro and in vivo potential of PSM to improve plant nutrition and growth [139]. The PSM-mediated bioavailable P is not utilized directly by plant or soil microbes; instead, it is quickly subject to precipitation or adsorption reactions in the immediate vicinity in which it is solubilized or desorbed by PSM [140]. Additionally, the P solubilization of exogenous PSM may be reduced due to the lack of persistence by competition with endogenous microbes for P resources, or by maladjustment of newly inoculated soil environment [141,142]. Hence, the PSM-enhanced P uptake from soil to plant likely occurs in the rhizosphere environment, which provides higher growth potential for PSM than bulk soils [74,142]. 

Rhizosphere PSMs, which are commonly considered as symbiotic or free–living microorganisms, are capable of colonizing rhizospheric plant roots, improving plant stress tolerance to drought, salinity, and heavy metals, and increasing the rhizospheric orthophosphate contents by inorganic or organic P solubilization [143,144,145]. Arbuscular mycorrhizal fungi (AMF) are the most common symbiont to increase P uptake capacity of plants [146]. Plants in symbiosis with AMF through formation of dense “cluster roots” can produce organic anions or H^+^ to release P_i_ from P minerals, enhancing soil P_i_ uptake by AMF and plants. The concentration of bioavailable P_i_ ions (e.g., H_2_PO_4_^–^) in plants and AMF cells can reach about 1000-fold higher than in the soil solution [147]. Using a compartmented pot system with an isotope ^33^P labeled pool dilution, the P uptake performance of wheat was significantly improved by high levels of AMF *Rhizophagus irregularis* colonization from soluble P, dried sewage sludge, and incinerated sewage sludge [74]. Moreover, AMF can enhance plant P uptake by recruiting and enriching beneficial microbes, including soil PSM, in the extensive hyphae under nutrient-scarce conditions, and thus provide a wider physical exploration of P undepleted soil [148]. 

Nevertheless, the ability of AMF to acquire P from the soil P_o_ pool may depend on the reactions with rhizospheric PSM because several AMF lack NSAPs genes [149]. Hence, the beneficial interactions between AMF and PSM occur by providing key resources for each other [150]. AMF can extend hyphae and transform PSM to the P_o_ pool. The exudates (e.g., sugars, carboxylates, amino acids) released by AMF or plants then stimulate the growth of rhizospheric PSM, and further improve the P_o_ mineralization, which, in turn, gives positive effects on the uptake of P by AMF and plants (Figure 4) [151]. A similar study found that the fructose excreted by AMF *Rhizophagus irregularis* stimulated the expression of phosphatase genes in PSB *Rahnella aquatilis*. Meanwhile, the fructose stimulated the release rate of phosphatase by regulating the protein secretary system (Figure 4), promoting the P_i_ release from P_o_ mineralization and the subsequent P_i_ uptake by AMF [152]. Using metagenomics and amplicon sequencing, increasing microbial communities with P solubilizing abilities and soil P cycling potentials are found in the hyphae-associated communities, also suggesting that AMF can recruit rhizospheric PSM to transfer P-containing nutrients from AMF hyphae to their symbiotic plants and enhance the P uptake from soil to plants [153,154]. 

## 9. Conclusions and Future Prospects

PSMs have gained wide acceptance as environmentally friendly and easily acquired P fertilizers to improve soil orthophosphate concentrations and P geochemical cycles since the beginning of the 20th century. A considerable number of PSMs with P_o_ or P_i_ solubilizing characteristics have been isolated, with selected ones used in plant growth and soil trails. These PSMs are highly abundant in soil and definitely have important roles in the global biogeochemistry of soil P cycling, including mineralization, solubilization, desorption, dissolution, and weathering. However, large scale in situ field tests are still urgently required. Previous studies have discussed improvements in plant growth, such as biomass increase, rhizosphere expansion, and physiological properties, while changes are usually ignored in important intermediates or metabolites of plants and PSMs, plant signal transductions, P morphological variations, and plant-microbe interactions during P_o_ or P_i_ solubilization processes in the rhizosphere. For example, although AMF are obligate biotrophs that compensate the host plants through nutrient acquisition and produce extensive extraradical hyphae as habitat for other soil microbes, they can only utilize P_o_ and improve plant P uptake in symbiosis with PSMs, as they lack the ability to secrete phosphatases [150,152]. Unfortunately, the relative contribution of PSMs to the acquisition of P in this symbiotic relationship remains largely unknown.

P cycling in soil depends on a combination of physical, chemical, and biological processes. Inoculation with PSM—which are always non-indigenous—as P fertilizers, may be considered a biological intervention or anthropogenic perturbation affecting the soil P cycle and native microbial community structure. It is a great hurdle to characterize the growth and the functions of PSM in soil. Recent studies have introduced some novel approaches (e.g., isotope labeling, metagenomics) to discuss the resistance of inoculated PSMs and their roles in soil P cycle [155]. Liang et al. [25] found that previously unknown PSB isolates were capable of solubilizing P_o_ and played important roles in driving the enhancement of soil P cycle using metagenomic sequencing. Hong et al. recently pointed out Raman spectroscopy as a promising tool for identifying microbial phenotypic and functional heterogeneity at the single-cell level without destroying the original cells or samples. Li et al. [156] applied single-cell Raman spectroscopy coupled with D_2_O labeling to probing P_o_ and P_i_ solubilizing bacteria, which discerned and located PSB in a mixed bacterial medium and complex soil communities at the single-cell level. These approaches may give us more novel options to understand the ecological functions and risks of PSM inoculation in soil P migration behavior, spatiotemporal variation characteristics of biologically available P, and soil indigenous microbial communities. In addition, studies on spatiotemporal changes in soil P dynamics and the relative rates of C and N erosion are required to improve our understanding of the roles of PSMs in the soil P cycle, and to develop some conceptual C and N biogeochemistry models for dynamic landscapes [129,131]. Based on the contribution of PSMs to global P cycling, there are important ecological, biogeochemical, and financial reasons to improve our understanding on PSM abilities and their utilization potentials in agricultural development.

## Figures and Tables

**Figure 1 biology-10-00158-f001:**
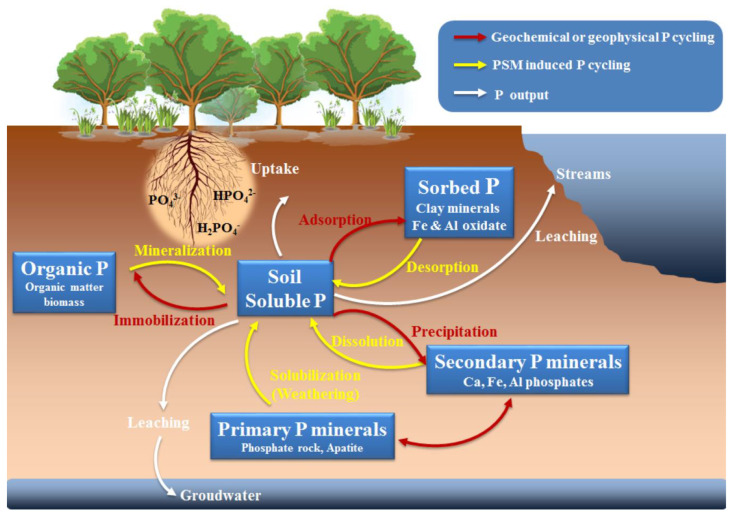
A schematic diagram of soil phosphorus (P) biogeochemical cycles. The red arrows indicate geochemical or geophysical P reactions and cycles. The yellow arrows indicate phosphate solubilizing microorganisms (PSM)-induced P reactions and cycles. The white arrows indicate P flows linking plants, streams, and groundwater.

**Figure 2 biology-10-00158-f002:**
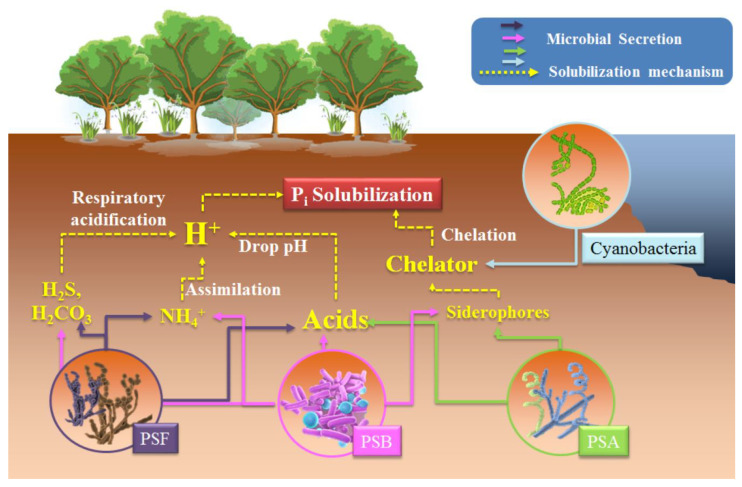
A schematic diagram of possible inorganic P (P_i_) solubilization mechanisms in phosphate solubilizing microorganisms (PSM). Arrows of different colors indicate the possible excretion agents by phosphate solubilizing fungi (PSF, purple), phosphate solubilizing bacteria (PSB, pink), phosphate solubilizing actinomycetes (PSA, aqua), and cyanobacteria (cyan), respectively.

**Figure 3 biology-10-00158-f003:**
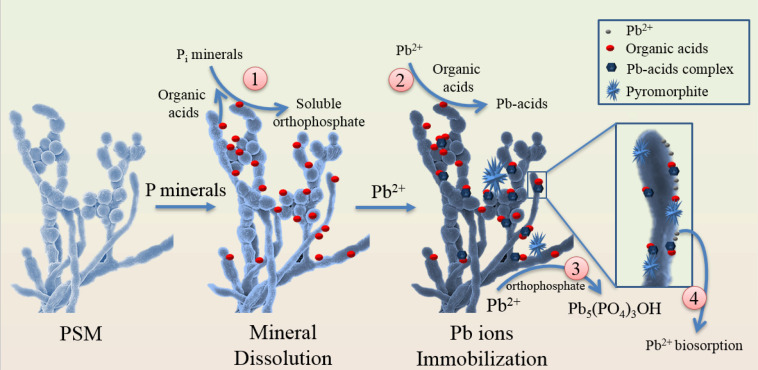
Simplified representation of phosphate solubilizing microorganisms (PSM)-induced dissolution to accelerate lead (Pb) precipitation to form secondary inorganic P (P_i_) minerals. ① PSM can dissolve Pi minerals to soluble orthophosphate by releasing organic acids. ② Pb ions in the solution are precipitated to Pb–acid complexes by reacting with organic acids that are released by PSM. ③ In the presence of orthophosphate (H_2_PO_4_^–^, HPO_4_^2–^), Pb ions are precipitated to the relatively stable pyromorphite. ④ Pb ions in the solution are biosorbed to PSM surface due to the negative charges of the surface functional groups.

**Figure 4 biology-10-00158-f004:**
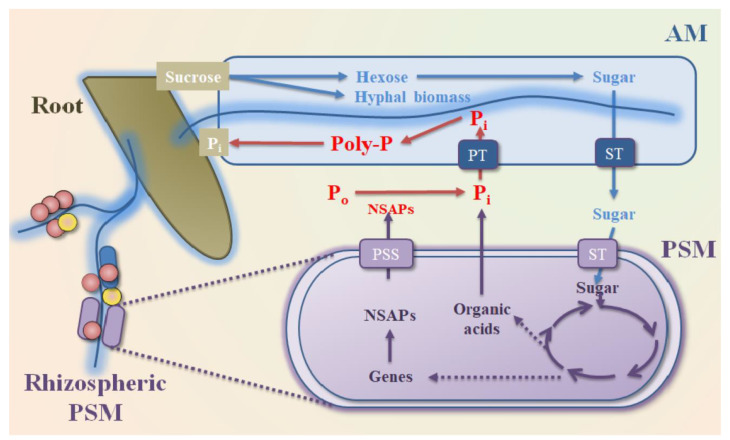
A schematic diagram of possible phosphorus (P) and sugar transport in roots, arbuscular mycorrhizal (AM), and phosphate solubilizing microorganisms (PSM) [151,152]. Arrows of different color indicate the possible reaction locations of roots (brown), AM (blue), and PSM (purple). Letters of different color indicate the possible sugar reactions (blue), P cycle (red), and microbial metabolism of PSM (purple), respectively. P_i_ for inorganic P, P_o_ for organic P, ST for sugar transporter, PSS for protein secretory system, and NSAPs for non-specific acid phosphatases.

## Data Availability

Not applicable.

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
