# Peer review of "Roles of Phosphate Solubilizing Microorganisms from Managing Soil Phosphorus Deficiency to Mediating Biogeochemical P Cycle"

_biology, 2021, doi:10.3390/biology10020158_

Round 1
Reviewer 1 Report
The manuscript by well written and organised. I don’t have any real concern with the structure and the information contained in review. However, I don’t find any innovation in it. The topic has been reviewed several times and even book has been published quite recently Phosphate-Solubilizing Microorganisms: A Critical Review (2015). Furthermore, the review is quite didactic, thus it could be a good book chapter, but I regret to say that does not provide either an innovative, nor a critical point of view on the topic. The only part that to my knowledge was quite original is chapter 6 “PSM–induced dissolution in accelerating metal precipitation to form secondary Pi minerals”
My suggestion is to restructure it to provide an advancement beyond the revie already published on the topic.
I included minor suggestions direcly in the attached PDF

Author Response
February 8, 2021
Dear Assistant Editor Fancy Yan and Reviewers:
Thank you very much for your letter commenting the manuscript entitled “Roles of phosphate solubilizing microorganisms from managing soil phosphorus deficiency to mediating biogeochemical P cycle” (biology-1090728). We highly appreciate your consideration and the constructive comments on our manuscript from all the Reviewers. We have carefully revised our manuscript to address all the comments and made correction to each comment which we hope it can meet with the requirements of editor. Incorrect writings are revised in the paper. The point-by-point list of our responses is presented below.
Thanks to the Reviewers for questioning the innovation of this manuscript, and their significant suggestions to perfect this manuscript. The P solubilization mechanisms and plant growth promotions of PSM have been largely and systematically discussed in previous Reviews. In this Review, we are focusing on the important roles of PSM in soil biogeochemical P cycle, as much less study has been discussed about the variations of P cycle and availabilities in response of PSM participation. As the Reviewers suggested, the original manuscript may make the readers understand the PSM instead of their roles in P cycle, resulting in the innovation reduction of this manuscript. So, we have extensively revised and reconstructed this manuscript. We have removed Table, reconstructed a paragraph and added a new schematic diagram, shortened some discussions that were fully discussed in previous Reviews, and cited more recently published papers to introduce more recent findings about PSM.
We also asked Prof. Fei Ge, Prof. Dayi Zhang, the co-authors of this work, to revise the final version and the responses to the comments. All the grammar and syntax error changes in the revised manuscript will not influence the content and the framework of the paper. All the revisions, including grammar and syntax errors, tables and figures, and references are revised as marked in the Manuscript-Marked revisions.
Special thanks to Assistant Editor Fancy Yan for reading our work so patiently and warming the work earnestly, and all the Reviewers’ suggestions to perfect our manuscript. We hope all these structural changes and revisions could reach the Reviewer’s suggestions and requests.
Once again, thank you very much for your letter and comments.
Sincerely
Pro.f Xingwang Liu
Department of Environmental Science and Engineering
Xiangtan University
Xiangtan, Hunan 411105, China
E-mail: lxwsa@xtu.edu.cn
-----------------------------------------------------------------------------------------------------
All changes made to the text are showing as follows so that they may be easily identified, and all the comments passed by the reviewer are answered below. The main correction in the paper and the responds to the reviewer’ comments are as flowing:
Response to Reviewer 1:
Comment 1. The manuscript by well written and organised. I don’t have any real concern with the structure and the information contained in review. However, I don’t find any innovation in it. The topic has been reviewed several times and even book has been published quite recently Phosphate-Solubilizing Microorganisms: A Critical Review (2015). Furthermore, the review is quite didactic, thus it could be a good book chapter, but I regret to say that does not provide either an innovative, nor a critical point of view on the topic. The only part that to my knowledge was quite original is chapter 6 “PSM–induced dissolution in accelerating metal precipitation to form secondary Pi minerals”
My suggestion is to restructure it to provide an advancement beyond the revie already published on the topic.
Response: Thank very much for the reviewer’s questioning about the innovation of this manuscript, and the significant suggestions to revise this manuscript. As the Reviewer suggested, it is not a new topic about the phosphate solubilizing microorganisms (PSM). From the early Review reports (for example, Hilda Rodrı́gueza and Reynaldo Fraga, Phosphate solubilizing bacteria and their role in plant growth promotion, 1999), the Book (Mohammad Saghir Khan, et al, Phosphate Solubilizing Microorganisms, 2014), to the latest Reviews (Pratibha Rawat, et al, Phosphate-Solubilizing Microorganisms: Mechanism and Their Role in Phosphate Solubilization and Uptake, 2020) and Book Chapters (Estimation of Phosphate Solubilizing Capacity of Microorganisms, 2021), PSM have been largely and systematically discussed. My research group has great interests in PSM since 2011. After thorough and comprehensive readings, we found that most of the previous books and published research papers and reviews mainly discussed the inorganic and organic phosphate (P) solubilizing ability, the P solubilizing mechanism and plant growth promotion of PSM, which focusing in distinguishing PSM’ s contributions to plant nutrition acquisition, and understanding how the soil P is released by microorganisms to enhance soil P availability and improve soil health. While, a comprehensive and complete understanding about the roles of PSM in P geochemical processes has not received the merited attentions.
In this Review, we are focusing on the important roles of PSM in soil biogeochemical P cycle, as much less study has been known about the variations of P cycle and availabilities in response of PSM participation. As the Reviewer suggested, the original manuscript may make the readers understand the PSM instead of their roles in P cycle, resulting in the innovation reduction of this manuscript. So, we have reconstructed this manuscript in following four parts. Other revisions about writings are revised as marked in the Manuscript-Marked revisions.
Part 1.
As the Reviewer suggested, we have removed the Table 1, in which the PSMs and microbial secretions were commonly discussed in many previous reviews and book chapters, in the original manuscript to reduce the discussions about the PSMs and their P solubilizing mechanisms. The mineralization of organic P and the solubilization of inorganic P are the two main P biogeochemical processes that PSMs are involved in soil, so we suggest to keep these two chapters, and the Figure 2. Meanwhile, we have re-named the title “3. PSM enhance soil P cycle through organic P mineralization”, and rewritten the paragraph from line 163-172 as “Hence, manipulation of the orthophosphate release from soil Po sources is an important soil P cycle, which has the potential to increase the availability of soil Po for plant uptake and reduce reliance on the P fertilizer inputs. Soil microbes, especially PSM, can enhance soil Po cycle through Po mineralization and de-composition. By analyzing soil P pools and oxygen isotope ratios in P (δ18OP), Bi et al. have uncovered that soil microbes could activate the soil P cycle by promoting extracellular hydrolysis of Po compounds and facilitating the turnover of bioavailable P pools (H2O-Pi, NaHCO3-Pi and NaOH-Pi). These biogeochemical processes are mainly moderated by the activities of phosphatase enzymes in PSM and soils”, and line 185 to 187 as “Although soil PSM involve in Po mineralization and P cycle at various scales, the dominant enzymes and the functional genes are always similar” to provide an advancement on the mineralization biogeochemical process.
Part 2.
We have shortened Part 4 and wiped off the secondary titles “4.1 Pi solubilizing microflora” and “4.2 Pi solubilizing mechanisms”, and re-named the title “4. PSM–mediated inorganic P solubilization to enhance soil orthophosphate contents”. Pi solubilizing microorganisms and their solubilizing mechanisms are fully discussed in other previous published Reviews and book chapters. So similar structures, research results and conclusions are inevitably discussed in this part. In the revised manuscript, we have not listed all the recent isolated stains. Instead, we are focusing on describing the commonly isolated PSMs that are possible responsible for regulating soil P cycles, and told the readers that bacteria, fungi, actinomycetes and cyanobacteria are commonly identified as PSM. In the Pi solubilizing mechanisms, we are focusing on the discussions about the microbial secretions to solubilize insoluble P compounds in Fig.2. Moreover, we introduced several examples to explain the possible Pi solubilizing mechanisms by chemistry (from line 266 to 278) as “The carboxyl groups of organic acids can bind P by replacing cations or compete for P adsorption sites, enhancing the soil absorption of PO43– and increasing Pi solubilization. Notable levels of organic acid production and Pi solubilization performance are achieved by PSM isolates. Although PSB represent the largest PSM population in soil environment, PSF exhibit greater Pi solubilization abilities by producing 10 times more organic acids than PSB, declining pH by 1–2 units in both liquid and solid medias. Furthermore, different Pi minerals show a range of H+ production and Pi solubility levels, which can be explained based on Ksp values, acidity coefficients and chemical equilibria. In strong acid conditions, PO34− will be dissolved first from Pi minerals and protonated to form hydrogen P (HPO42− or H2PO4−). Metal ions (e.g., Ca2+, Fe3+ or Al3+) are likely to subsequently capture the hydrogen P to form metal hydrogen P with generally higher Ksp values than their equivalent metal P”, which were newly discussed in recent studies (Luyckx 2020, Jiang 2020).
Part 3.
We have re-constructed this manuscript by add a new chapter “8. PSM enhance P uptake from soil to plant in rhizosphere environment”, which was “4.3 soluble Pi release and plant growth promotion” in the original manuscript. The plant growth promotion is another commonly discussed content in previous published Reviews and book chapters. This manuscript is focusing on the P cycle in soil. So as the Reviewer suggested, we have rewritten this part from line 463 to 510. In this part, we did not discuss the plant growth promotions of PSM, instead, we demonstrated the P cycle from soil to plant, which is always an important P cycle in soil from Pi to Po. Although PSM are well-known factor to enhance P uptake and plant growth, their mechanisms to improve soil Pi content and plant uptake are rarely discussed. Moreover, the PSM-mediated bioavailable P is not utilized directly by plant or soil microbes; instead, it is quickly subject to precipitation or adsorption reactions in the im-mediate vicinity in which it is solubilized or desorbed by PSM. So, we have rewritten this part to newly discuss the functions of AMF with P solubilization performance, and the possible genetic and protein mechanisms by reviewing some recently published papers. Meanwhile, we have drawn a new figure (Fig. 4) from these papers to make these mechanisms more accessible to readers.
As we have known so far, no recent Reviews have discussed the Pi uptake enhancements by PSM from soil to plant. This rewritten part will also make this manuscript more innovative comparing to the similar Reviews.
Part 4.
In the last part Conclusions and future prospects, we have rewritten from line 538 to 551. In the original manuscript, we have settled many future research aspects about the ecology and bioinformation. After reading several new published research papers and reviews, we have found that research approaches to discuss the PSM are still timeworn. In order to improve our understanding about PSM and its role in P cycle, newly approaches are urgently needed. So we added some newly introduced methods to isolate PSM or demonstrate its eco-toxicity to soil environment. These revisions can inspire readers to use more recently innovated or commonly used in other aspects (e.g., chemistry, material, biology, geology).
After the four parts of structural revisions, this manuscript has been already major rewritten to produce some advancements and innovations comparing to other reviews. The language of the review has been revised by the co-authors, and the revied word or sentences are marked in the revied manuscript to make this manuscript more scientific and accurate. We hope all these structural changes could reach the Reviewer’s suggestions and request. Furthermore, we are highly appreciating for the Reviewer’s significant suggestions to perfect our manuscript.
Comment 2. Why future tense in line 17?
Response: Thank for reviewer’s good comment to perfect the manuscript. We have replaced to the inappropriate tense in line 17 in the revised manuscript.
Comment 3. Is the Simple Summary necessary?
Response: Thank for reviewer’s good comment to perfect the manuscript. Based on the formation of acceptable manuscript in Biology, it is necessary to keep this Simple Summary in the manuscript.
Comment 4. Change to “small” in line 24?
Response: Thank for reviewer’s good comment to perfect the manuscript. We have replaced to the “small” in line 24 in the revised manuscript.
Comment 5. Either there is accumulation, or a states in the next line lishviation..pls rephase?
Response: Thank for reviewer’s good comment to perfect the manuscript. We have replaced to the “inputs of soil P” to eliminate the misunderstandings from other readers.
Comment 6. Change to “terrestrial” in line 42?
Response: Thank for reviewer’s good comment to perfect the manuscript. We have replaced to the “terrestrial” in line 42 in the revised manuscript.
Comment 7. Change to “since” in line 48?
Response: Thank for reviewer’s good comment to perfect the manuscript. We have replaced to the “since” in line 48 in the revised manuscript.
Comment 8. What does the meaning of “because of their abnormal”?
Response: Thank for reviewer’s good comment to perfect the manuscript. It means the P can keep at the normal liquid and solid states instead of the gaseous phase at earth temperatures and pressures. In order to eliminate the misunderstandings from other readers, we have changed the sentence to “In ecosystems, the P cycle differs from the N and C biogeochemical cycles since it does not form any stable gaseous species at Earth temperatures and atmospheric pressures” from line 47 to 49 in the revised manuscript.
Comment 9. Despite several atmospheric transport exists as explained in the text, thus this paragraph needs to be clarified?
Response: Thank for reviewer’s good comment to perfect the manuscript. we have changed the sentence to “In ecosystems, the P cycle differs from the N and C biogeochemical cycles since it does not form any stable gaseous species at Earth temperatures and atmospheric pressures” from line 47 to 49 in the revised manuscript.
Comment 10. This concept has been repeated from line 127 to 128c ?
Response: Thank for reviewer’s good comment to perfect the manuscript. We have eliminated this repeated sentence from 127 to 128 in the revised manuscript.
Comment 11. What is the “piggybacks” means in line 130?
Response: Thank for reviewer’s good comment to perfect the manuscript. This word is a miswriting. In order to preface this sentence, we have rewritten “Pi exists in different forms and proportions in soil, which may leach into streams to deposit P in ocean sediments and to slowly reshape the earth’s surface in primary Pi cycle, or be taken up by plants or soil microorganisms in the secondary Po cycle” from line 123 to 126 in the revised manuscript.
Comment 12. Acronyms also in full in the cations?
Response: Thank for reviewer’s good comment to perfect the manuscript. We have added the full explanations of all the acronyms in every figure caption in the revised manuscript.
Comment 13. In terms of cell numbers? Biomass? Frequence in line 232?
Response: Thank for reviewer’s good comment to perfect the manuscript. We have replaced to the “form the largest microbial communities with P solubilization abilities in soil” from line 233 to 234 in the revised manuscript.
Comment 14. Replace to strong in line 239?
Response: Thank for reviewer’s good comment to perfect the manuscript. We have replaced to the “strong” in line 241 in the revised manuscript.
Comment 15. Replace to “can be beneficial to” in line 242?
Response: Thank for reviewer’s good comment to perfect the manuscript. We have replaced to the “can be beneficial to” in line 244 in the revised manuscript.

Reviewer 2 Report
The purpose of the review is to lay the foundations for future understandings of the resource for research focused on the development of biochemical and genetic for phosphorus acquisition. The review, however, must be improved in terms of writing since some grammar and syntax errors are present in the manuscript. They should address the subject and critically review the information from the literature. There are several current works showing the effect of the phosphate solubilizing microorganisms mediate soil orthophosphate levels, and the biogenic mechanisms behind these activities. I suggest updating the references with current references. There are several other responses mitigated by phosphate solubilizing microorganisms in plants that were not mentioned in the review. Authors should make a great effort to leave the review complete.
Introduction sections need more convincing rational for this review.
Line 151: “3. Role of PSM in organic P mineralization and soil P cycling”. The important thing is that all heterotrophic and aerobic bacteria and fungi produce a variety of organic acids from carbon sources through the biochemical pathway of tricarboxylic acid, so it is not important to identify the type of acid. On the other hand, different phosphorus sources can affect the production of acids only quantitatively and are not effective in the profile of acids. I hope the authors understand what I mean.
Lines 198 – 207: At the beginning of the discussion, better explore the possibilities of using metagenomics, in prospecting for new phytases of biotechnological interest. Read and quote: Farias, Nathálya; Almeida, Isabela; Meneses, Carlos. 2018. "New Bacterial Phytase through Metagenomic Prospection" Molecules 23, no. 2: 448. https://doi.org/10.3390/molecules23020448
Line 224: “4.1 Pi solubilizing microflora”. There have been several studies in past reporting phosphate solubilization and growth promotion of plants. The authors presented similar reports as the main conclusion of their review. The identification of bacterial strains needs more detailed analysis including phylogenetic analysis.
Line 312: “4.3 Soluble Pi release and plant growth promotion”. The main limitation of the review is normally phosphate solubilizing microorganisms is a symbiotic organisms and may be able to solubilize insoluble phosphates. Authors fail to explain about the mechanism of solubilization except carrying some plant growth promoting activities. The presented may not be sufficient enough to prove the role of phosphate solubilizing microorganisms for plant growth promotion. Also there were no informations related to nitrogen content and also no history of phosphate solubilizing microorganisms strains used in this review. Read and quote: German A. Estrada-Bonilla, Ademir Durrer, Elke J.B.N. Cardoso, Use of compost and phosphate-solubilizing bacteria affect sugarcane mineral nutrition, phosphorus availability, and the soil bacterial community, Applied Soil Ecology, Volume 157, 2021, 103760, ISSN 0929-1393, https://doi.org/10.1016/j.apsoil.2020.103760. Romero-Perdomo F, Beltrán I, Mendoza-Labrador J, Estrada-Bonilla G and Bonilla R (2021) Phosphorus Nutrition and Growth of Cotton Plants Inoculated With Growth-Promoting Bacteria Under Low Phosphate Availability. Front. Sustain. Food Syst. 4:618425. doi: 10.3389/fsufs.2020.618425
Author Response
February 8, 2021
Dear Assistant Editor Fancy Yan and Reviewers:
Thank you very much for your letter commenting the manuscript entitled “Roles of phosphate solubilizing microorganisms from managing soil phosphorus deficiency to mediating biogeochemical P cycle” (biology-1090728). We highly appreciate your consideration and the constructive comments on our manuscript from all the Reviewers. We have carefully revised our manuscript to address all the comments and made correction to each comment which we hope it can meet with the requirements of editor. Incorrect writings are revised in the paper. The point-by-point list of our responses is presented below.
Thanks to the Reviewers for questioning the innovation of this manuscript, and their significant suggestions to perfect this manuscript. The P solubilization mechanisms and plant growth promotions of PSM have been largely and systematically discussed in previous Reviews. In this Review, we are focusing on the important roles of PSM in soil biogeochemical P cycle, as much less study has been discussed about the variations of P cycle and availabilities in response of PSM participation. As the Reviewers suggested, the original manuscript may make the readers understand the PSM instead of their roles in P cycle, resulting in the innovation reduction of this manuscript. So, we have extensively revised and reconstructed this manuscript. We have removed Table, reconstructed a paragraph and added a new schematic diagram, shortened some discussions that were fully discussed in previous Reviews, and cited more recently published papers to introduce more recent findings about PSM.
We also asked Prof. Fei Ge, Prof. Dayi Zhang, the co-authors of this work, to revise the final version and the responses to the comments. All the grammar and syntax error changes in the revised manuscript will not influence the content and the framework of the paper. All the revisions, including grammar and syntax errors, tables and figures, and references are revised as marked in the Manuscript-Marked revisions.
Special thanks to Assistant Editor Fancy Yan for reading our work so patiently and warming the work earnestly, and all the Reviewers’ suggestions to perfect our manuscript. We hope all these structural changes and revisions could reach the Reviewer’s suggestions and requests.
Once again, thank you very much for your letter and comments.
Sincerely
Pro.f Xingwang Liu
Department of Environmental Science and Engineering
Xiangtan University
Xiangtan, Hunan 411105, China
E-mail: lxwsa@xtu.edu.cn
-----------------------------------------------------------------------------------------------------
All changes made to the text are showing as follows so that they may be easily identified, and all the comments passed by the reviewer are answered below. The main correction in the paper and the responds to the reviewer’ comments are as flowing:
Response to Reviewer 2:
Comment 1. The purpose of the review is to lay the foundations for future understandings of the resource for research focused on the development of biochemical and genetic for phosphorus acquisition. The review, however, must be improved in terms of writing since some grammar and syntax errors are present in the manuscript. They should address the subject and critically review the information from the literature. There are several current works showing the effect of the phosphate solubilizing microorganisms mediate soil orthophosphate levels, and the biogenic mechanisms behind these activities. I suggest updating the references with current references. There are several other responses mitigated by phosphate solubilizing microorganisms in plants that were not mentioned in the review. Authors should make a great effort to leave the review complete.
Introduction sections need more convincing rational for this review.
Response: Thanks very much for the reviewer’s significant suggestions to perfect this manuscript. As the Reviewer suggested, it is not a new topic about the phosphate solubilizing microorganisms (PSM). From the early Review reports (for example, Hilda Rodrı́gueza and Reynaldo Fraga, Phosphate solubilizing bacteria and their role in plant growth promotion, 1999), the Book (Mohammad Saghir Khan, et al, Phosphate Solubilizing Microorganisms, 2014), to the latest Reviews (Pratibha Rawat, et al, Phosphate-Solubilizing Microorganisms: Mechanism and Their Role in Phosphate Solubilization and Uptake, 2020) and Book Chapters (Estimation of Phosphate Solubilizing Capacity of Microorganisms, 2021), PSM have been largely and systematically discussed. After thorough and comprehensive readings, we found that most of previous books and published research papers and reviews mainly discussed the inorganic and organic phosphate (P) solubilizing ability, the P solubilizing mechanism and plant growth promotion of PSM, focusing in distinguishing PSMs’ contributions to plant nutrition acquisition, and understanding how the soil P is released by microorganisms to enhance soil P availability with the aim of restoring other soil geochemical and improving soil health. While, a comprehensive and complete understanding about the roles of PSM in P geochemical processes has not received the merited attentions. So, in this Review, we are focusing on the important roles of PSM in soil biogeochemical P cycle, as much less study has been known about the variations of P cycle and availabilities in response of PSM participation.
As the Reviewer suggested, this manuscript has been already major rewritten to increase advancements and innovations comparing to other reviews. We have added several advanced research results and recent publications to improve the innovation of this manuscript. The newly cited publications are marked red in the Reference in the revised manuscript. Moreover, we have rewritten the part of PSM enhanced Pi uptake by plants as Reviewer suggested. The language of the manuscript has been revised by the co-authors, and the revised word or sentences are marked in the revied manuscript to make this manuscript more scientific and accurate. We hope all these structural changes could reach the Reviewer’s suggestions and request. Furthermore, we are highly appreciating for the Reviewer’s significant suggestions to perfect our manuscript.
Comment 2. Line 151: “3. Role of PSM in organic P mineralization and soil P cycling”. The important thing is that all heterotrophic and aerobic bacteria and fungi produce a variety of organic acids from carbon sources through the biochemical pathway of tricarboxylic acid, so it is not important to identify the type of acid. On the other hand, different phosphorus sources can affect the production of acids only quantitatively and are not effective in the profile of acids. I hope the authors understand what I mean.
Response: Thanks very much for the reviewer’s significant suggestions to perfect this manuscript. As the Reviewer suggested, different P sources, including organic and inorganic sources, will affect PSM to produce different organic acids. The acids, chelator, and exopolysaccharide and other microbial secretions that are responsible for P solubilizations, were commonly discussed in many previous reviews and book chapters. So in this manuscript, which focusing on the P cycles in soil, it is not necessary to list the acids by different PSM stains in response to different P sources. So, we have removed the Table 1 in the original manuscript to reduce the discussions about the PSMs and their P solubilizing mechanisms. Meanwhile, we have shortened the discussions about the enzymes in Po mineralization and acids in Pi solubilization mechanisms. Instead, we added several recently published papers to discuss the newly results about PSM in Po mineralization or Pi solubilization from line 163 to 172, line 209 to 218, line 550 to 563 in the revised manuscript.
Comment 3. Lines 198 – 207: At the beginning of the discussion, better explore the possibilities of using metagenomics, in prospecting for new phytases of biotechnological interest. Read and quote: Farias, Nathálya; Almeida, Isabela; Meneses, Carlos. 2018. "New Bacterial Phytase through Metagenomic Prospection" Molecules 23, no. 2: 448. https://doi.org/10.3390/molecules23020448
Response: Thanks very much for the reviewer’s significant suggestions to perfect this manuscript. As the Reviewer suggested, we have accepted and cited the publications that Reviewer suggested, and have rewritten the sentences from line 209 to 218 as “While, a great diversity of phytase is existed in the vast majority of unculturable soil microorganisms, which have been rarely studied. Using metagenomics, Farias et al. constructed environmental genomic libraries to determine the completely sequencing of the clone phytase gene from unculturable soil micro-organisms in red rice crop residues and castor bean cake. The newly isolated phytase enzyme showed high hydrolase activity at neutral pH under β-propeller structure. Therefore, it is crucial to develop and utilize more advanced approaches to support the roles of PSM–derived enzymes in releasing free ortho-phosphate from organic P forms in the soil P cycle”. The cited publication is marked in the revised manuscript.
Meanwhile, we added the discussions about the new approaches to characterize the role of PSM in environment from line 545 to 558 as “It is a great hurdle to characterize the growth and the functions of PSM in soil envi-ronment. Recent studies have introduced some novel approaches (e.g. isotope labeling, metagenomics) to discuss the resistance of inoculated PSMs and their roles in soil P cy-cle [155]. Liang et al. [25] found that previously unknown PSB isolates were capable of solubilizing Po and played important roles in driving the enhancement of soil P cycle using metagenomic sequencing. Hong et al. recently pointed out Raman spectroscopy as a promising tool for identifying microbial phenotypic and functional heterogeneity at the single-cell level without destructing the original cells or samples. Li et al. [156] firstly applied that single-cell Raman spectroscopy coupled with D2O labeling to prob-ing Po and Pi solubilizing bacteria, which discerned and located PSB in a mixed bacte-rial medium and complex soil communities at the single-cell level. These approaches may give us more novel options to understand the ecological functions and risks of PSM inoculation in soil P migration behavior, spatiotemporal variation characteristics of biologically available P, and soil indigenous microbial communities”. We have added several advanced and recently published research papers to demonstrate the importance of utilizing now approaches in PSM studies. The newly cited publications are marked in the revised manuscript.
Comment 4. Line 224: “4.1 Pi solubilizing microflora”. There have been several studies in past reporting phosphate solubilization and growth promotion of plants. The authors presented similar reports as the main conclusion of their review. The identification of bacterial strains needs more detailed analysis including phylogenetic analysis.
Response: Thanks very much for the reviewer’s significant suggestions to perfect this manuscript. As the Reviewer noted, Pi solubilizing microorganisms and their solubilizing mechanisms are fully discussed in other previous published Reviews and book chapters. So similar structures, research results and conclusions are inevitably discussed in this part. We have shortened Part 4 and wiped off the secondary titles “4.1 Pi solubilizing microflora” and “4.2 Pi solubilizing mechanisms”, and re-named the title “4. PSM–mediated inorganic P solubilization to enhance soil orthophosphate contents”. In the revised manuscript, we have not listed all the recent isolated stains. Instead, we are focusing on describing the commonly isolated PSMs that are possible responsible for regulating soil P cycles, and told the readers that bacteria, fungi, actinomycetes and cyanobacteria are commonly identified as PSM. In the Pi solubilizing mechanisms, we are focusing on the discussions about the microbial secretions to solubilize insoluble P compounds in Fig.2.
Comment 5. Line 312: “4.3 Soluble Pi release and plant growth promotion”. The main limitation of the review is normally phosphate solubilizing microorganisms is a symbiotic organisms and may be able to solubilize insoluble phosphates. Authors fail to explain about the mechanism of solubilization except carrying some plant growth promoting activities. The presented may not be sufficient enough to prove the role of phosphate solubilizing microorganisms for plant growth promotion. Also there were no informations related to nitrogen content and also no history of phosphate solubilizing microorganisms strains used in this review. Read and quote: German A. Estrada-Bonilla, Ademir Durrer, Elke J.B.N. Cardoso, Use of compost and phosphate-solubilizing bacteria affect sugarcane mineral nutrition, phosphorus availability, and the soil bacterial community, Applied Soil Ecology, Volume 157, 2021, 103760, ISSN 0929-1393, https://doi.org/10.1016/j.apsoil.2020.103760. Romero-Perdomo F, Beltrán I, Mendoza-Labrador J, Estrada-Bonilla G and Bonilla R (2021) Phosphorus Nutrition and Growth of Cotton Plants Inoculated With Growth-Promoting Bacteria Under Low Phosphate Availability. Front. Sustain. Food Syst. 4:618425. doi: 10.3389/fsufs.2020.618425
Response: Thanks very much for the reviewer’s significant suggestions to perfect this manuscript. As the Reviewer noted, the 4.3 part“Soluble Pi release and plant growth promotion” in line 312 in original manuscript was not sufficient enough to prove the role of phosphate solubilizing microorganisms for plant growth promotion. The plant growth promotion is commonly discussed content in previous published Reviews and book chapters. Several researchers have found that PSM could improve the plant growth without enhancing soil orthophosphate concentrations. Moreover, it is not suitable to discuss the plant growth promotion or plant nitrogen uptake in this manuscript that is focusing on soil P cycle.
As the Reviewer suggested, we have read all the suggested publications, and cited in the manuscript from line 473 to 489 as “Bioavailable P content in soil is an important factor to enhance plant P uptake and achieve higher crop yields [72,138]. Soil PSM can employ different biogeo-chemical strategies to make use of the unavailable P forms and in turn help in enhancing P uptake from soil to plants. Hence, most studies have considered PSM as a promising inoculant/biofertilizer for raising the productivity of agro-nomic crops in agroecological niches [72,138,139]. However, the soil is a more diverse and spatially heterogenic matrix than growth medium, which will result in some of the discrepancies between the in vitro and in vivo potential of PSM to improve plant nutrition and growth [140]. The PSM-mediated bioavailable P is not utilized directly by plant or soil microbes; instead, it is quickly subject to precipitation or adsorption reactions in the immediate vicinity in which it is solubilized or desorbed by PSM [141]. Additionally, the P solubilization of ex-ogenous PSM may be reduced due to the lack of persistence by competition with endogenous microbes for P resources, or by maladjustment of newly inoculated soil environment [142,143]. Hence, the PSM-enhanced P uptake from soil to plant likely occurs in rhizosphere environment which provides higher growth potential for PSM than bulk soils [74,143]”.
Meanwhile, we have re-constructed this manuscript by adding a new chapter “8. PSM enhance P uptake from soil to plant in rhizosphere environment”, which was “4.3 soluble Pi release and plant growth promotion” in the original manuscript. This manuscript is focusing on the P cycle in soil. So as the Reviewer suggested, we have rewritten this part from line 463 to 510. In this part, we did not discuss the plant growth promotions of PSM, instead, we demonstrated the P cycle from soil to plant, which is always an important P cycle in soil from Pi to Po. Although PSM are well-known factor to enhance P uptake and plant growth, their mechanisms to improve soil Pi content and plant uptake are rarely discussed. Moreover, the PSM-mediated bioavailable P is not utilized directly by plant or soil microbes; instead, it is quickly subject to precipitation or adsorption reactions in the im-mediate vicinity in which it is solubilized or desorbed by PSM. So, we have rewritten this part to newly discuss the functions of AMF with P solubilization performance, and the possible genetic and protein mechanisms by reviewing some recently published papers. As we have known so far, no recent Reviews have discussed the Pi uptake enhancements by PSM from soil to plant. This rewritten part will also make this manuscript more innovative comparing to the similar Reviews.
We have also eliminated the paragraph with nitrogen increment by PSM from line 334 to 350 in the original manuscript. As the Reviewer suggested, no information related to nitrogen content and no history of PSM are used in this manuscript. Moreover, this manuscript is discussing the importance of P cycle, the changes in N contents or cycles will make this manuscript lengthier and more difficult to read by other readers. So, we suggest to remove this discussion about nitrogen. Instead, we have added more information and genetic mechanisms about plant P uptake enhanced by PSM and AM from line 490 to 522. Meanwhile, we have drawn a new figure (Fig. 4) from these papers to make these mechanisms more accessible to readers.
We have marked all the changes of words, sentences, and references in the revised manuscript. We hope all these changes could reach the Reviewer’s suggestions and request. Furthermore, we are highly appreciating for the Reviewer’s significant suggestions to perfect our manuscript.

Round 2
Reviewer 1 Report
I think the manuscript has been substantially imporved and it's now ready for publication
Reviewer 2 Report
Thanks for attending all the suggestions. The Review has been significantly improved. In view of the above, I believe that the Review presents robust and consolidated content, bringing to light new information on roles of phosphate solubilizing microorganisms. I consider that the work has enough quality to be considered for publication in Biology (MDPI).